# Bumblebee Foraging Dynamics and Pollination Outcomes for Cherry Tomato and Pear Varieties in Northern China

**DOI:** 10.3390/insects15040216

**Published:** 2024-03-22

**Authors:** Xunbing Huang, Qianwen Zhang, Umer Ayyaz Aslam Sheikh, Yueyue Wang, Li Zheng

**Affiliations:** 1College of Resources and Environment, College of Agriculture and Forestry Science, Linyi University, Linyi 276000, China; xunbingh@163.com (X.H.); qianwenzhangsd@163.com (Q.Z.); 2Department of Entomology, Faculty of Agriculture, University of Poonch Rawalakot, Rawalakot 12350, Pakistan; umerayaz@upr.edu.pk; 3Key Laboratory of Natural Enemies Insects, Ministry of Agriculture and Rural Affairs, Jinan 250100, China

**Keywords:** bumblebee, pollination, foraging activities, fruit setting rate, production yield

## Abstract

**Simple Summary:**

Bumblebees are well-suited pollinators of economic crops. This study evaluated the foraging behavior and pollination effects of *Bombus terrestris* on cherry tomatoes and pears in northern China. *B. terrestris* pollination can improve cultivation efficiency, increase yield, and produce more economic benefits, which indicates that it has great potential in pollination applications for cherry tomatoes and pears in northern China.

**Abstract:**

Bumblebees (*Bombus terrestris*) have strong environmental adaptability and high pollen transfer efficiency, making them well-suited pollinators of economic crops. However, bumblebee pollination is still not widely applied in northern China due to the lack of data on foraging behavior and pollination effects. We conducted a three-year experiment involving cherry tomatoes (*Solanum lycopersicum* L.) and pears (*Pyrus* spp.) treated with bumblebee pollination to evaluate the foraging behavior and pollination effects on these two crops. Results showed that *B. terrestris* had enhanced foraging activities as daytime temperatures rose from 18 °C to 26 °C, as indicated by the increased number of bees leaving the hive and returning bees carrying pollen in greenhouses in winter. There were two peaks in the foraging activity of bumblebees in pear orchards in early spring, which was closely related to the temperature change in the daytime. Undoubtedly, cherry tomatoes treated with *B. terrestris* had higher fruit setting rate, weight, seed number, and fruit yields compared to those with hormone 2,4-dichlorophenoxyacetic acid treatments, as well as a lower rate of deformed fruits. *B. terrestris* pollination can significantly increase the fruit setting rate and fruit yield of pears, compared with open pollination, and can fully achieve the effect of hand pollination. *B. terrestris* pollination can improve cultivation efficiency, increase yield, and produce more economic benefits. Moreover, it can also contribute to reducing hormone residues and ensure the safety of agricultural products. We recommend its application to cherry tomatoes in greenhouses in winter and potential application to pears in orchards in early spring in northern China. However, the risk to local bumblebee species of introducing commercially available bumblebees into orchards should be considered and evaluated in future research. This study provides both empirical support and a theoretical basis for the selection of bumblebees as pollinators in the production of economically important crops and the improvement of crop cultivation management in northern China.

## 1. Introduction

Since Sladen’s pioneering proposal of bumblebee pollination application in crops in 1912, as documented in *The Bumblebee*, the utilization of bumblebees as pollinators has garnered substantial attention [1,2,3]. Bumblebees collect pollen and nectar to provide food resources and obtain essential amino acids, fats, sugars, vitamins, water, and other minerals for the survival and development of their population [4,5,6]. This collection behavior can help pollinate plants [7,8]. Recent studies have shown that bumblebees, as eusocial insects, have a very strong ability to learn and remember, and when food resources are scarce, they even can stimulate plants to bloom in advance by nibbling on plant leaves, inducing them to provide pollen and nectar [9].

*Bombus terrestris* is an important pollinator for wild plants and economically valuable crops. Owing to their strong capacity for pollination, they have been widely used in agricultural production in developed countries, especially in the Netherlands, Belgium, Israel, France, and other EU countries, where *Bombus terrestris* has been used to pollinate greenhouse tomatoes, peppers, eggplants, and other crops, producing a huge economic achievement [10]. For example, the use of *Bombus terrestris* pollination not only saves costs of labor and time to improve farming efficiency but also can improve the yield and quality of fruits and vegetables, reduce the occurrence of some plant diseases, and avoid hormone residues. These attributes are of great significance for the improvement of the economic benefits of farmers, the protection of the ecological environment, and ensuring the quality and safety of consumers’ food [11,12,13,14,15]. Bumblebees have important pollination utilization value, and the study of their behavior characteristics and effects on plants can provide a theoretical basis for their further application.

The widespread application of *Bombus terrestris* in the pollination of economically important crops is based on their excellent behavior characteristics. Scientists have conducted much research on related behavioral aspects of the bumblebee for over 100 years. For example, they have a long proboscis and a high tolerance for low temperature, low light, and high moisture [14,16,17]. They actively visit flowers when the temperature is 8 °C, while honeybees generally work above 16 °C [12]. Notably, bumblebees have the trait of “buzz pollination” and strong abilities for pollen collection and transfer. When pollinating tomatoes, worker bees can influence pollen release by adjusting the amplitude and frequency of wing vibration to obtain the most appropriate amount of pollen grains. The number of pollen grains foraging bumblebees gather can reach more than one million in a single collection [18,19,20]. In summer, some worker bees forage at 5:00 AM and stop working when it is dark [12]. Therefore, they have strong environmental adaptability and high pollen transfer efficiency, making them well-suited for the pollination of economically important crops.

CO_2_ can disrupt bumblebee diapause, marking a pivotal moment in overcoming diapause challenges and offering crucial technical support for the artificial mass breeding of bumblebees [21]. Subsequently, companies like Biobest in Belgium, and Koppert and Bunting Brinkman Bees (BBB) in the Netherlands, initiated bumblebee breeding and global popularization efforts [10]. Today, the utilization of *Bombus terrestris* for pollination has become a global practice, gaining recognition for its pivotal role. China is a hotspot for high bumblebee species richness. There are about 125 extant species, representing 14 of the 15 Bombus subgenera [8]. However, the domestication and artificial mass breeding of those native bumblebees have not succeeded. Hence, their reared populations are still not applied for the pollination of economic crops.

Despite the widespread use, bumblebee pollination technology is still applied less in China and has not been popularized in most areas. Farmers still mostly rely on traditional methods to improve fruiting rates and production yields. For example, in northern China, cherry tomatoes are mostly planted in autumn and summer in greenhouses to increase economic benefits [22]. The confined spaces in greenhouses limit tomato pollination due to a lack of natural pollinators, which could adversely affect fruit yield from extremely low fruiting rates. To improve tomato fruiting rate and production yields in greenhouses, the traditional method used most often was treating flowers using synthetic hormones, such as 2,4-dichlorophenoxyacetic acid (2,4-D) [15,23]. However, the excessive application of hormones or their residues can cause poisoning in humans or have adverse effects on the environment [7,24,25]. Furthermore, the use of synthetic hormones in treating flowers is expensive in terms of both labor and time. For the pollination of pear plants, farmers generally hand-pollinate using brushes during the flowering phase in early April, which is also expensive in terms of both labor and time. In some areas, farmers have even adopted open pollination with no hand pollination or bee releases in the north of China.

The underutilization of bumblebee pollination in northern China is primarily attributed to farmers’ lack of understanding of this technology and the existing knowledge gap regarding the foraging behavior and pollination effects of bumblebees on economically important crops in this region. To address this gap and facilitate the broader application of bumblebee pollination, comprehensive field studies comparing bumblebee pollination with other methods are essential. In this study, we aimed to evaluate the foraging behavior and pollination effects of *Bombus terrestris* on cherry tomatoes (*Solanum lycopersicum* L.) and pears (*Pyrus* spp.). Our objective is to provide empirical data and a theoretical foundation for selecting bumblebee pollination or alternative methods in the cultivation of economically important crops, thereby contributing to enhanced crop management in northern China.

## 2. Materials and Methods

### 2.1. Research Site

The research was conducted in Fei (35.318° N, 118.045° E), a large agricultural county, in Shandong Province, China. The mean annual temperature in the study area is 13.6 °C. Air temperatures can fall to −11 °C in December and reach 36 °C in July. The mean annual precipitation is 850 mm. Tomatoes and pears are important economic crops in this area. The annual tomato and pear production is up to ~270,000 tons and ~45,605 tons, worth USD 280 million and USD 23 million, respectively (http://nyj.linyi.gov.cn/) (accessed on 1 January 2023). To improve tomato production yields in greenhouses, the traditional method used most often was treating flowers by 2,4-D. For the pollination of pear plants in orchards, farmers generally hand-pollinate using brushes and even adopt open pollination. Bumblebee pollination technology is still less applied.

### 2.2. Bumblebee and Hormone Treatment for Cherry Tomatoes in Greenhouses

Although the use of bumblebees is an extremely effective method in some developed countries, tomato production in greenhouses is still overly dependent on the hormone 2,4-D in northern China. To reveal foraging behavior and ensure the pollination effects of bumblebees on cherry tomatoes in northern China, we conducted a three-year-long greenhouse experiment involving cherry tomatoes treated with bumblebees and the hormone 2,4-D.

The commonly cultivated cherry tomato variety (Fushan-88) in this region was planted in six greenhouses in September 2020, 2021, and 2022. The area of each greenhouse was 0.13 ha. All greenhouses were sealed from leaks and protected with insect nets to prevent pests from invading and bumblebees from escaping. Five thousand cherry tomato plants were planted in each greenhouse in rows spaced 40 cm apart with a plant spacing of 30 cm.

Cherry tomato flowers began to bloom in mid-October of each year. The duration of the flowering phase was about 45 days. At this time, the bumblebees were released to tomato plants in the three greenhouses. Each bumblebee hive, purchased from Shandong Lubao Technology Development Co., Ltd. (Jinan, China), had about 100 healthy bumblebees. One bumblebee hive was placed in each greenhouse until the end of the flowering phase. All bumblebee hive maintenance was conducted according to the instructions provided. Before installing them in the greenhouses, all three hives were kept in the laboratory room (temperature 26–28 °C; relative humidity 70%) for 24 h. All hives were comprehensively checked for health conditions and the presence of any pest or predator. Each hive was free from diseases and microbes. Bumblebee queens were glistening in color without any damage to their bodies and actively attended to the whole colony. About 100 active bumblebee workers with good-conditioned wings were present in the colony along with the founder queen. Each hive box held a plastic tank in the base of the box that stored a sugar solution as a substitute for nectar to sustain the nutritional development of bumblebee health, as tomato flowers do not produce nectar. With the interval of 7 days, a continual examination of colonies was conducted to check the population conditions and for the presence of pests or predators in the hive. Dead bumblebee workers were removed from the hive to avoid contamination.

Tomato plants in separate sets of the three greenhouses were treated with the hormone 2,4-D. During the flowering period, we used 5 mg × L^−1^ solution of 2,4-D to treat tomato inflorescence. About 1 mL of solution was used for each inflorescence. The same fertilization, irrigation practices, and fungicide treatments were applied to all six greenhouses. No insecticides were used, and the fungicides applied had no documented insecticidal activity.

### 2.3. Monitoring of Temperature, Relative Humidity, and Bumblebee Foraging Behavior in Cherry Tomato Greenhouses

To monitor the temperature and relative humidity in greenhouses with bumblebee releasing, a HOBO U23 Pro v2 Temperature/Relative Humidity Data Logger (Onset Computer Corporation, Bourne, MA, USA) was installed in each greenhouse. The heat-retaining quilts covering the greenhouses of cherry tomatoes were opened at 9:00 in November 2021. We monitored bumblebee foraging activities from 9:00 to 17:00 in each greenhouse with bumblebee release on 29 November 2020. In each greenhouse, we made hourly observations and recorded the number of bees leaving the bumblebee hive within 10 min (individuals per 10 min), as well as the number of returning bees carrying pollen within 10 min (individuals per 10 min) from 9:00 to 17:00. At the same time, ten bumblebees leaving the bumblebee hive were randomly selected to observe the number of visited flowers per min per individual, as well as the single-flower residence time (s) of bumblebees during the period from 12:00 to 13:00.

### 2.4. Investigation of the Effect of Bumblebee and Hormone Treatments on Cherry Tomatoes

In each of the six greenhouses, one hundred tomato flowers bitten by bumblebees (with obvious bite marks) or the flowers of twenty tomato plants treated with 2,4-D were randomly selected and marked with strings in 2020 to observe the development. The number of flowers, fruit, and malformed fruit of each selected plant was monitored and recorded to calculate the fruit set rate (%) given by the number of fruits on the tomato plant/the number of flowers and the malformed fruit rate (%) given by the number of malformed fruit/the number of fruits. After full red ripening, fifty fruits from these marked flowers were randomly collected and weighed (g) in each of the greenhouses. Then, each fruit was dissected to record the number of seeds. After tomato harvest, the tomato fruit yield (kg per 0.13 ha) in the six greenhouses was recorded in 2020, 2021, and 2022 to compare cherry tomato production for bumblebee and hormone treatments.

### 2.5. Bumblebee Pollination, Hand Pollination, and Open Pollination in Pear Orchards

To reveal the foraging behavior and ensure the pollination effects of bumblebees on pears in northern China, we conducted a two-year orchard experiment involving pears treated with bumblebee pollination, hand pollination, and open pollination.

Nine pear orchards in the research area were selected to conduct the pollination experiment in early spring in 2021 and 2022. The area of each orchard was 0.1 ha, and the distance between each orchard was >5 km. The commonly cultivated pear varieties “Qiuyue” and “Huangguan” were planted in each of the nine pear orchards for seven years. Each orchard included 80 “Qiuyue” pear plants and 10 “Huangguan” pear plants as pollinizers. The same fertilization, irrigation practices, and fungicide treatments were applied to all orchards. No insecticides were used, and the fungicides applied had no documented insecticidal activity at the pear flowering phase.

Pear flowers began to bloom in early April of each year. At this time, the bumblebees were released to the pear plants in three of the nine orchards. One bumblebee hive was placed in the center of each orchard until the end of the flowering phase. The bumblebee pollination of pear flowers lasted ~14 days in each orchard. The condition and maintenance of each bumblebee hive was the same as above. Another three orchards were randomly selected to adopt hand pollination. Here, the farmers hand-pollinated flowers using brushes dipping with pollen to touch the pear flowers during the flowering phase in early April. The pollen of the pear variety “Huangguan” was used for hand pollination. The remaining three orchards adopted open pollination, with no hand pollination or reared bumblebees introduced to the area. The observed natural pollinators in pear orchards mainly included insects occurring in early spring, such as wild honeybees.

### 2.6. Monitoring of Temperature, Relative Humidity, and Bumblebee Foraging Behavior in Pear Orchards

To monitor the temperature and relative humidity in orchards with bumblebee releases, the HOBO Logger, as described above, was also installed in bumblebee-pollinated orchards. We conducted the monitoring of bumblebee foraging activities from 6:00 to 18:00 h in each orchard with bumblebee release on 6 April 2021. In each orchard, we conducted hourly observations and recorded the number of bees leaving the bumblebee hive and the number of returning bees carrying pollen within 10 min from 6:00 to 18:00 h. In each orchard, ten bumblebees leaving the bumblebee hive were randomly selected to observe the number of visited flowers per min per individual and the single-flower residence time (s) during the same period.

### 2.7. Investigation of the Effects of the Three Pollination Methods on Pears

In each of the nine orchards, ten trees were randomly selected (~50 m apart) to survey the effects of different pollination methods in 2021. Flower counts in each tree were performed on six lateral branches selected from the top (two branches), middle (two branches), and bottom (two branches) of the main stem. These selected branches were marked with strings. After pollination treatment, the number of fruits and malformed fruits of each selected tree was monitored to calculate the fruit set rate (%) and the malformed fruit rate (%) using the same method described above. After full ripening, eighteen pear fruits from these marked branches (three fruits per branch) in each tree were randomly collected and weighed (g). After the pear harvest, the pear fruit yield (kg per 0.1 ha) in the nine orchards was recorded in 2021 and 2022 to compare pear fruit production for the three pollination methods.

### 2.8. Data Analysis

The Student’s *t*-test was used to compare the fruit set rate, malformed fruit rate, fruit weight, and fruit yield of cherry tomatoes treated by bumblebees and hormones. One-way analysis of variance (ANOVA) and Tukey’s HSD were used to compare the above variables of pears treated using three pollination methods. All tests were conducted using SAS version 9.0 after verifying the normality.

## 3. Results

### 3.1. Bumblebee Foraging Activities in Greenhouses of Cherry Tomatoes

The heat-retaining quilts covering the greenhouses of cherry tomatoes were opened at 9:00 in November 2020. Temperature and relative humidity monitoring (Figure 1A) in greenhouses of cherry tomatoes, from 9:00 to 17:00 h, showed that the overall trend of temperature changes rose from ~18 °C to a maximum value of ~26 °C at 12:30 and then decreased to ~18 °C at 17:00 PM. The relative humidity decreased from 62% to a minimum of 49% at 13:00 and then gradually increased to 54% by 17:00.

The monitoring of bumblebee foraging activities (Figure 1B) showed that the number of bees leaving the bumblebee hive (individuals per 10 min) increased to ~12 individuals from 9:00 to 12:00 h and then gradually decreased to zero by 17:00. Similarly, the number of returning bees carrying pollen (individuals per 10 min) first increased to ~9 individuals from 9:00 to 12:30 h and then gradually decreased to zero by 17:00.

### 3.2. Bumblebee Foraging Activities in Pear Orchards

Temperature and relative humidity monitoring (Figure 2A) in pear orchards from 6:00 to 17:00 h showed that the overall trend of temperature changes rose from ~14 °C to a maximum of ~30 °C at 13:00 and then decreased to ~19 °C by 18:00. The relative humidity decreased from 58% to a minimum of 27% at 15:00 and then gradually increased to 36% by 18:00.

The monitoring of bumblebee foraging activities (Figure 2B) showed that the number of bees leaving the bumblebee hive (individuals per 10 min) exhibited two peak periods. During the first peak period, the number of bees leaving bumblebee hive increased to ~15 individuals from 9:00 to 10:00 h and then gradually decreased to 2 by 13:00. During the second peak period, the number of bees leaving the bumblebee hive increased to ~12 individuals from 13:00 to 15:00 h and then gradually decreased to zero by 18:00. Similarly, the number of returning bees carrying pollen (individuals per 10 min) also exhibited two peak periods. During the first peak period, the number of returning bees carrying pollen increased to ~11 individuals from 9:00 to 11:00 and then gradually decreased to 2 by 14:00. During the second peak period, the number of returning bees carrying pollen increased to ~9 individuals from 14:00 to 16:00 and then gradually decreased to zero by 18:00.

### 3.3. Foraging Behavior of Bumblebees Visiting Flowers

The monitoring of bumblebee behaviors (Table 1) showed that the average number of visited flowers of a single bumblebee per min for cherry tomatoes and pears was 8.6 and 6.4, respectively. The average single-flower residence time of bumblebees on cherry tomatoes and pears was 5.6 s and 7.1 s, respectively.

### 3.4. Effects of Bumblebee Pollination on Cherry Tomato Fruit

The performance of cherry tomato plants in response to bumblebee pollination was compared to hormone (2,4-D) treatment (Figure 3). The results showed that cherry tomato plants pollinated by bumblebees exhibited a significantly higher fruit setting rate (96.13%, Figure 3A, *t* = 71.283, *p* < 0.001), fruit weight (12.43 g, Figure 3C, *t* = 9.506, *p* < 0.05) and number of seeds (106.67, Figure 3D, *t* = 56.862, *p* < 0.001), compared to those with hormone treatment (87.75%, 10.65 g, 43.21). In contrast, bumblebee-pollinated tomato plants had a significantly lower deformed rate of fruit (3.56%, Figure 1B, *t* = 14.636, *p* < 0.01) than those treated with hormone 2,4-D (6.47%). The fruit setting rate, fruit weight, and seed amounts of bumblebee-pollinated tomatoes increased by 9.55%, 16.71%, and 146.86%, respectively, with the deformed rate of fruit decreasing by 44.98%.

### 3.5. Effects of Bumblebee Pollination on Pear Fruit

The performance of pear plants in response to bumblebee pollination, hand pollination, and open pollination were evaluated and compared (Figure 4). The results showed that the fruit setting rate of pear plants treated with bumblebee pollination (78.45%) and hand pollination (74.19%) was significantly higher compared to open pollination (54.13%) (Figure 4A) (*F* = 71.247, *p* < 0.05). The fruit-setting rate increased by 44.93% and 37.06%, respectively. There were no significant differences in the fruit weight and deformed rate of fruit between pear plants treated using the three pollination methods (Figure 4B,C).

### 3.6. Yield Effects of Bumblebee Pollination on Cherry Tomatoes

The fruit yield of cherry tomatoes pollinated by bumblebees in greenhouses was recorded to compare with the fruit yield of those subjected to hormone treatment (Figure 5). The results showed that cherry tomato plants pollinated by bumblebees exhibited a significantly higher fruit yield in 2021 (7725.36 kg per 0.13 ha, *t* = 17.233, *p* < 0.05), 2022 (7860.52 kg per 0.13 ha, *t* = 29.6, *p* < 0.01), and 2023 (7569.79 kg per 0.13 ha, *t* = 11.068, *p* < 0.05), compared to those with hormone treatment (7257.33 kg per 0.13 ha; 7370.68 kg per 0.13 ha; 7268.51 kg per 0.13 ha), respectively. The fruit yield increased by 6.45%, 6.65%, and 4.14%, respectively, with an average increase of 5.75%.

### 3.7. Yield Effects of Bumblebee Pollination on Pears

The fruit yield of pear plants using bumblebee pollination, hand pollination, and open pollination was recorded and compared in 2021 and 2022 (Figure 6). The results showed that the fruit yield of pear plants treated with bumblebee pollination (2021: 3006.33 kg per 0.1 ha, 2022: 3134.67 kg per 0.1 ha) and hand pollination (2021: 2942.86 kg per 0.1 ha, 2022: 3095.63 kg per 0.1 ha) were significantly higher (*F* = 9.416 and 7.591, *p* < 0.05) compared to natural pollination (2021: 2590.67 kg per 0.1 ha, 2022: 2825.82 kg per 0.1 ha). The fruit yield increased by 16.04% and 13.59% in 2021 and by 10.93% and 8.72% in 2022, respectively. There were no significant differences in fruit yield between pear plants treated using bumblebee pollination and hand pollination.

## 4. Discussion

Temperature is an important environmental factor affecting the foraging activities of bumblebees [17]. Although bumblebees have strong environmental adaptability, such as high tolerance to low temperature, low light, and high moisture, and weak phototaxis, their foraging activities were also restricted within a certain temperature range. Too low or too high a temperature is not conducive to bumblebee foraging during the day [7,16]. In our monitoring experiments in cherry tomato greenhouses, we found that bumblebees had enhanced foraging activities when the daytime temperature rose from 18 °C to 26 °C, indicated by the increased number of bees leaving the bumblebee hive and returning bees carrying pollen. When the temperature dropped from 26 °C in greenhouses, the foraging activities of bumblebees were reduced. The temperature rise within greenhouses in winter in northern China was conducive to bumblebee foraging activities during the daytime.

During early spring in pear orchards, bumblebees exhibited different foraging activities compared with the performance in cherry tomato greenhouses in winter. We found that there were two peaks in the daytime foraging activity of bumblebees, which was closely related to the temperature change from 6:00 to 18:00 h. Bumblebees began to forage at 6:00 (temperature ~14 °C) in the morning and reached the first activity peak with the increased temperature. However, the number of foraging activities of bumblebees decreased rapidly from 11:00 to 13:00 h, while the temperature in this period was close to 30 °C, which may be unfavorable to the foraging activities of bumblebees. Afterward, the temperature dropped to ~25 °C, at which time the second peak of foraging activity occurred, and the foraging activity stopped before dark. Temperatures can directly influence bumblebee foraging activities and also indirectly affect them by influencing the flowering state of plants. Some studies have found that temperature can affect the anther cracking and pollen release of plant flowers [7,26]. With the increase in daytime temperature and the decrease in relative humidity in the orchard, the anther cracking of fruit trees will increase and a large amount of pollen will be released, providing a sufficient honey and pollen source for bumblebees and inducing them to forage and therefore exert their pollination role. Detailed studies to better understand the relationship between bumblebees, temperature, and relative humidity are needed in the future.

The monitoring of bumblebee behaviors showed that the number of visited flowers per min and the single-flower residence time of bumblebees for cherry tomatoes and pears were different. Many factors, including biotic or abiotic factors, can influence these foraging behaviors. Among them, the foraging behavior of bumblebees was closely related to plant species, especially for the physical or chemical traits of flowers, such as the plant flower color and taste, the stage of flowering time, the nutrient level, or volatile organic compounds [27,28,29,30,31,32,33,34,35,36,37,38]. Additional laboratory studies should be conducted to unravel the mechanism underlying this observation.

After long-term co-evolution, bumblebees, as pollinators, have a strong ability to identify the activity of plant pollen, select and collect pollen with strong activity and good quality, and visit flowers at the best time when the pollen is ripe, which all contribute to the improvement of the crop fruit setting rate and yield [11,12,19,38]. Undoubtedly, cherry tomatoes treated with bumblebees had higher fruit yields compared to those with hormone treatment and exhibited a higher fruit setting rate, weight, and seed number, as well as a lower rate of deformed fruits. These results are consistent with similar findings from previous research on the effects of bumblebees on common tomatoes in greenhouses [11,15,39]. In contrast, the use of 2,4-D is not a natural pollination process; therefore, it may have resulted in less seed fruit and decreased fruit weight [15,39]. Additionally, the excessive use of 2,4-D could leave residues in the environment, the accumulation of which could be potentially poisonous to humans or other non-target organisms [25]. The use of reared bumblebees can reduce the levels of hormone residues in fruits or the environment. As effective pollinators, bumblebee pollination significantly improved tomato quality and production compared to traditional methods, which is worthy of wide application in tomato cultivation in greenhouses during winter in northern China.

In pear orchards, bumblebee pollination can significantly increase the fruit setting rate and fruit yield compared with open pollination, and it can fully achieve the effect of hand pollination. However, we should recognize that bumblebee pollination can save costs of labor and time investment to improve cultivation efficiency when compared with hand pollination. In addition, due to the relatively short flowering period of pears, after pear pollination, the reared bumblebees can continue to be used to pollinate other economically important crops, such as apples and melons, to exert their pollination value and produce additional economic benefits. As effective pollinators, reared bumblebee pollination can improve cultivation efficiency, which is worthy of application to pear cultivation in orchards in early spring in northern China. However, in the case of experiments involving hand pollination and bumblebee pollination in the present study, the tested pear tree branches were not covered with a net, so the results were the synergic effect of open pollination. We also acknowledged that the number of observations (three per group) for the total yield of pears was small in this study. More ingenious experiments and observations should be conducted to ensure the pollination effects are measured exactly in future research.

Factors such as climate change, habitat change, agricultural practices, etc., can affect bumblebee populations and their pollination applications [40]. For example, food scarcity due to habitat destruction caused by abnormally high temperatures and excessive farming practices could significantly reduce wild bumblebee populations [41]. It has been found that the increase in landscape heterogeneity contributed to the increase in the wild bumblebee population [42]. Therefore, more protective measures should be taken in the process of agricultural production to promote an increase in the wild bumblebee population. For example, we can plant a large number of flowering plants in spring and summer in areas where bumblebees are distributed to provide sufficient pollen sources to meet the food needs of the wild bumblebee population [43]. Moreover, in greenhouses, unreasonable farm operations will have a significant impact on reared bumblebee pollination applications. For example, abnormally high or low temperatures caused by unreasonable temperature and humidity control in greenhouses will cause the death of reared bumblebees or the decline of their pollination ability. The escape of bumblebees caused by a broken plastic membrane was not repaired in time and led to the decline of reared bumblebees [10]. Hence, farmers should constantly conduct optimal farming operations to protect and improve the pollination of bumblebees.

In addition, the irrational use of chemical pesticides by farmers is an important factor affecting the bumblebee population and pollination applications. Research has shown that the unreasonable use of neonicotinoid chemical pesticides, such as imidacloprid, thiamethoxam, and thiamethoxam have lethal and sublethal effects on bumblebees [44,45]. Neonicotinoid insecticides can inhibit the brain development of bumblebees and destroy the normal function of their nervous system, thus severely affecting their learning and memory ability and preventing bumblebees from normal foraging, returning to the nest, and reproducing [46,47]. Therefore, when using bumblebees for pollination in agricultural production, the effects of chemical pesticides on bumblebees should be fully considered to ensure their safety [48]. In recent years, with the development of biological pest control technology, bumblebee pollination technology has been combined with natural enemies and microbial pesticides to achieve a win–win situation between crop pollination and pest control. For example, while using bumblebees to pollinate tomatoes, we can release *Encarsia formosa* Gahan, *Orius sauteri* Poppius, *Asaphes vulgaris* Walker, and *Aphidoletes abietis* Kieffer and other natural enemies to control the whiteflies *Trialeurodes vaporariorum* West-wood, *Bemisia tabaci* Gennadius, *Lipaphis erysimi* Kaltenbach, and *Thrips flevas* Schrank [14,17].

Bumblebee pollination, as an important emerging technology in north China, can improve crop quality, increase yield, and produce huge economic benefits. It can also play an important role in regulating agricultural operations, restricting the use of chemical pesticides, reducing hormone residues, and ensuring the safety of agricultural products. With the increasing planting area of economically important crops [49], bumblebee pollination has great potential in China. In the future, we should further strengthen the research, promotion, and application of bumblebee pollination technology in China.

However, the risk of introducing commercially available bumblebees into orchards in northern China should be considered and evaluated cautiously. The global success of *B. terrestris* as both a commercial pollinator and an invasive species has unfortunate impacts. Evidence has exhibited that this species could negatively impact other bumblebees, even leading to local extirpations of native competitors as it continues to spread [50,51]. In addition, researchers have highlighted the reared *B. terrestris* as a risk factor for wild bumblebees due to the transfer of pathogens [52,53,54]. Thus, there should be an intense awareness of commercially available bumblebees to decrease the risk of interaction with wild bumblebee populations. Moreover, domestication and the artificial mass breeding of native bumblebees for pollination are the best approaches to avoid the risk of introducing non-native bumblebees.

## Figures and Tables

**Figure 1 insects-15-00216-f001:**
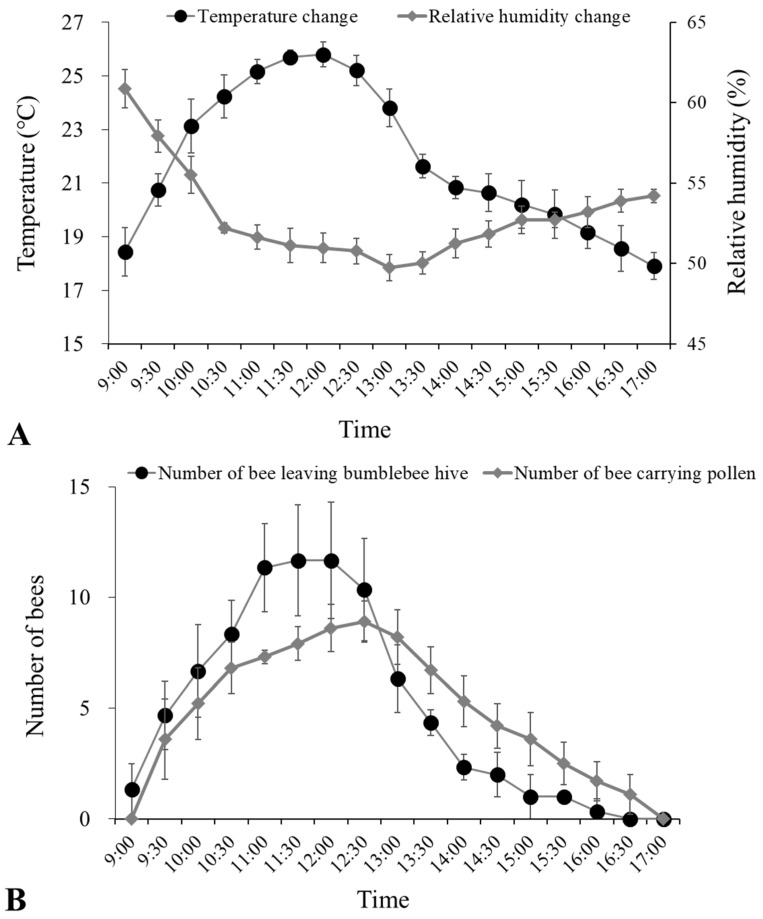
(**A**) The change of temperature (°C) and relative humidity (%), and (**B**) the number of bees leaving the bumblebee hive and returning bees carrying pollen (individuals per 10 min) in greenhouses of cherry tomatoes from 9:00 to 17:00 on 29 November 2020.

**Figure 2 insects-15-00216-f002:**
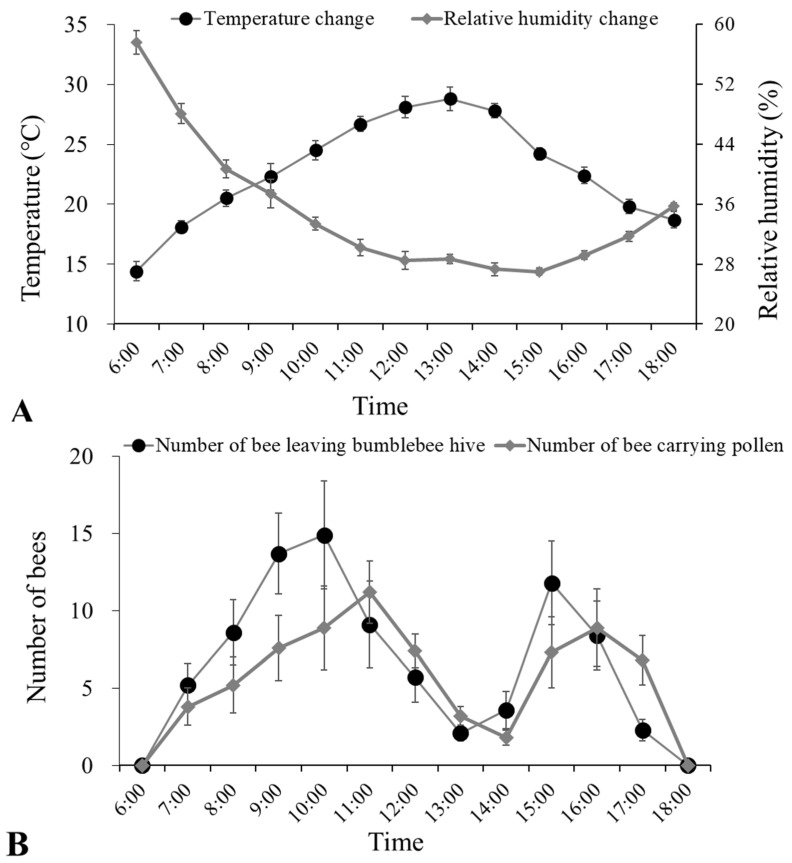
(**A**) The change of temperature (°C) and relative humidity (%), and (**B**) the number of bees leaving the bumblebee hive and returning bees carrying pollen (individuals per 10 min) in pear orchards from 6:00 to 18:00 h on 6 April 2021.

**Figure 3 insects-15-00216-f003:**
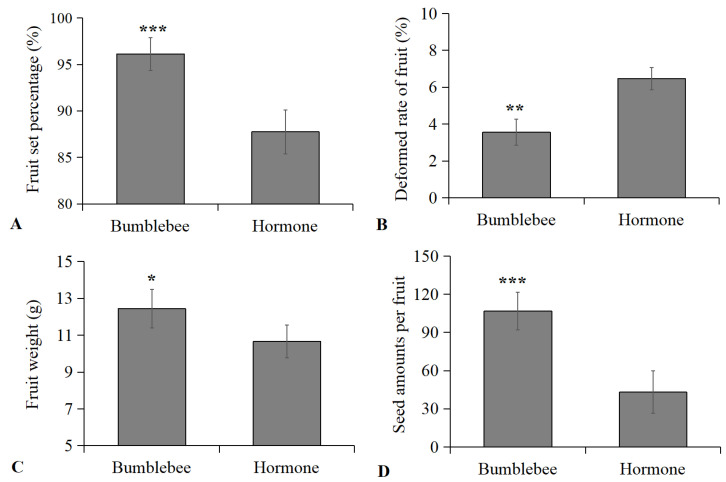
(**A**) Fruit setting rate (% ± SD), (**B**) deformed rate of fruit (% ± SD), (**C**) fruit weight (g ± SD), and (**D**) seed amounts per fruit (±SD) of cherry tomatoes treated with bumblebee and hormone 2,4-D. * *p* < 0.05, ** *p* < 0.01, *** *p* < 0.001 (Student’s *t* test).

**Figure 4 insects-15-00216-f004:**
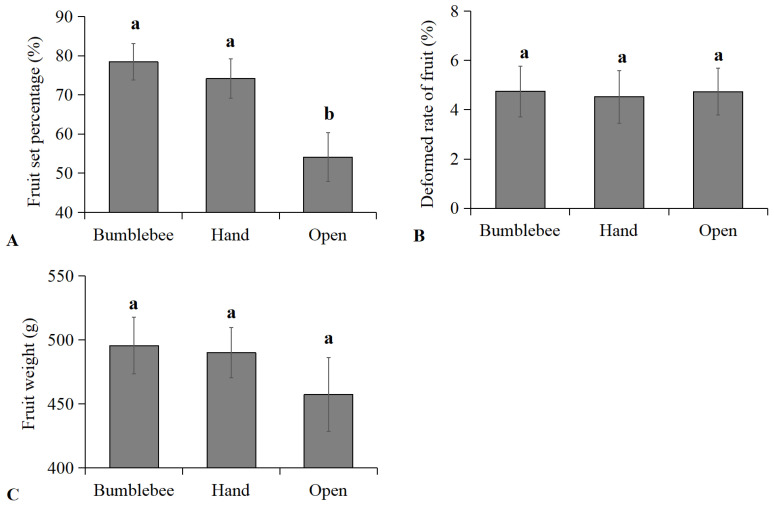
(**A**) Fruit setting rate (% ± SD), (**B**) deformed rate of fruit (% ± SD) and (**C**) fruit weight (g ± SD) of pears using bumblebee pollination, hand pollination and open pollination. Significant differences between three pollination methods are indicated by lowercase letters (ANOVA, Tukey’s HSD analysis, *p* < 0.05).

**Figure 5 insects-15-00216-f005:**
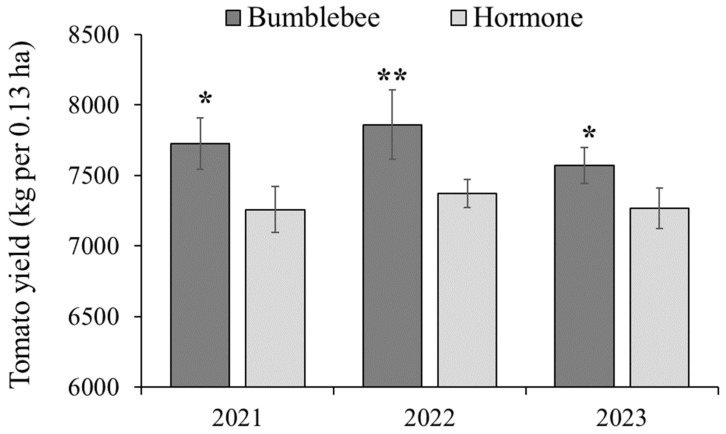
Production yield (kg ± SD per 0.13 ha) of cherry tomatoes treated with bumblebees and the hormone 2,4-D in 2020, 2021 and 2023. * *p* < 0.05, ** *p* < 0.01 (Student’s *t*-test).

**Figure 6 insects-15-00216-f006:**
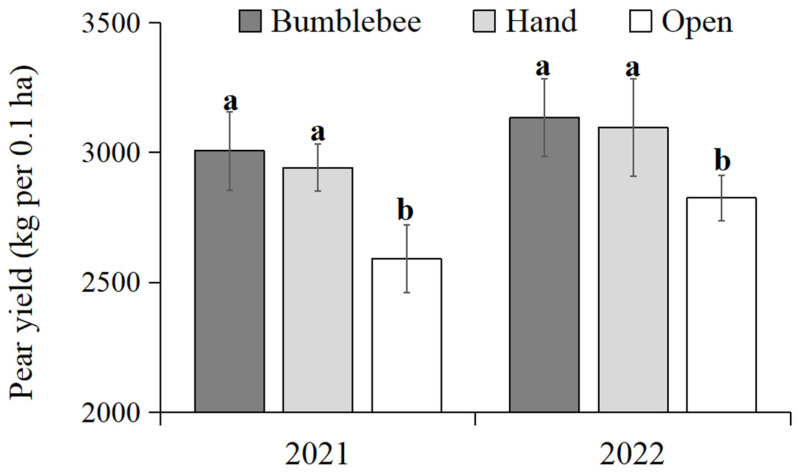
Production yield (kg ± SD per 0.1 ha) of pears using bumblebee pollination, hand pollination and open pollination in 2021 and 2022. Significant differences between three pollination methods are indicated by lowercase letters (ANOVA, Tukey’s HSD analysis, *p* < 0.05).

**Table 1 insects-15-00216-t001:** Foraging behavior of bumblebee flower visits on cherry tomatoes and pears.

Flowers	Number of Visited Flowers per min	Single-Flower Residence Time (s)
Cherry tomato	8.6 ± 1.2	5.6 ± 0.7
Pear	6.4 ± 0.9	7.1 ± 0.5

## Data Availability

The data that support the findings of this study are available from the corresponding author upon reasonable request.

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
