# Peer review of "Bumblebee Foraging Dynamics and Pollination Outcomes for Cherry Tomato and Pear Varieties in Northern China"

_insects, 2024, doi:10.3390/insects15040216_

Round 1

Reviewer 1 Report

Comments and Suggestions for Authors

The study has an applied purpose comparing the efficiency of hand pollination or hormonal regulation and bumblebee pollination. One species of vegetable and one fruit tree were tested, a tomato in greenhouse cultivation and a pear tree in field cultivation. The result confirms the high efficiency of bumblebees in pollination of tomatoes. For pear trees, it has not been determined what is more beneficial pollination by hand or by bumblebees 

The manuscript fulfils the formal requirements and is prepared and written carefully, but some explanations of the study methods and other issues are necessary. 

The introduction lacked information about the tested plants, especially their pollination requirements. Instead of in the introduction, a few pieces of information were in the M&M section (lines 104-106; lines 18-110). 

I suggest using the terminology adequate to the study of pollination and commonly used. 

Instead of natural pollination, use open pollination, and artificial pollination, use hand pollination. 

Beehive means space for the honeybee. I suggest using a bumblebee hive or bumblebee colony. 

In methods, the fruit set and deformed rates are not clearly defined (lines 145-148). In addition, it is difficult to verify whether the fruit set of the pear was controlled for the initial fruit before the fall of the initial fruit set or for the ripe harvested fruits.  

In the case of hand pollination, the tested tree branches were not covered with a net, so the results were the effect of hand pollination and open pollination. 

What cultivar was the pollen used for hand pollination?  

If the orchard was a single variety (bumblebee pollination), without pollinizer and the pollen used in the hand pollination was another variety, it is difficult to compare pollination methods 

Was the distribution of variables tested before selecting statistical parametric testsOn the other hand, the number of observations for the total yield (3 per group) was too small to compare averages.  What was treated as an observation when the average percentage of fruit set was compared? Greenhouse/orchard or plant/tree? 

Line 371-375 The idea is great, but it applies to plants that bloom later than pear trees because, at that time in nature, the bumblebee queens are only active, and the colonies are in the early stages of development. 

Line 374. Plant flowers do not secrete honey but nectar, however, bumblebees primarily need pollen. 

When interpreting results (e.g., lines 366-381), it is important to distinguish between facts that apply to populations of wild bumblebees vs. reared bumblebees 

In the discussion, the authors should make a clearer distinction between the use of bumblebees in the greenhouse versus in the field. Also, nothing mentioned about the relationship of commercial bumblebee colonies vs. wild bumblebees and wild bees. As well as there is a lack of interpretation of pollination results and reference to literature data. 

The reference list is double-numbered.

Author Response

Mar. 14, 2024

Dear reviewer,

We now return the revised edition of Manuscript ID insects-2893366, titled "Bumblebee foraging dynamics and pollination outcomes for cherry tomato and pear varieties in northern China".

We carefully revised the manuscript according to your comments, and addressed each comment (please see below), and believe that the manuscript has been improved.

We are grateful to your constructive comments on earlier drafts of this manuscript.

Thanks

With kind regards,

Yueyue Wang

Below we address each your comments:    
(1) The study has an applied purpose comparing the efficiency of hand pollination or hormonal regulation and bumblebee pollination. One species of vegetable and one fruit tree were tested, a tomato in greenhouse cultivation and a pear tree in field cultivation. The result confirms the high efficiency of bumblebees in pollination of tomatoes. For pear trees, it has not been determined what is more beneficial pollination by hand or by bumblebees. The manuscript fulfils the formal requirements and is prepared and written carefully, but some explanations of the study methods and other issues are necessary. The introduction lacked information about the tested plants, especially their pollination requirements. Instead of in the introduction, a few pieces of information were in the M&M section (lines 104-106; lines 18-110). 

Response: Yes, as the reviewer said, in pear orchards, bumblebee pollination can significantly increase fruit setting rate and fruit yield, compared with natural pollination, and can fully achieve the effect of artificial pollination. However, we should recognize that bumblebee pollination can save costs of labor and time investment to improve cultivation efficiency, compared with artificial pollination. We have discussed this in the Discussion section.

The pollination requirements of tested plants were provided the introduction section in the revised manuscript. In northern China, cherry tomatoes are mostly planted in autumn and summer in greenhouses to increase economic benefits. The confined spaces in greenhouses limit tomato pollination due to a lack of natural pollinators, which could adversely affect fruit yield from extremely low fruiting rates. To improve tomato fruiting rate and production yields in greenhouses, the traditional method used most often was treating flowers using synthetic hormones, such as 2,4-dichlorophenoxyacetic acid (2,4-D). However, the excessive application of hormone or its residues can cause poisoning in humans or have adverse effects on the environment. And, the use of synthetic hormones treating flowers are expensive in terms of both labor and time. For pollination of pear plants, farmers generally hand-pollinate using brushes during the flowering phase in early April, also expensive in terms of both labor and time. In some areas, farmers even adopted natural pollination with no artificial pollination or bee releases in the north of China. Thanks for your suggestions.

(2) I suggest using the terminology adequate to the study of pollination and commonly used. Instead of natural pollination, use open pollination, and artificial pollination, use hand pollination. Beehive means space for the honeybee. I suggest using a bumblebee hive or bumblebee colony. 

Response: Have corrected these according to your suggestions in the revised manuscript. Thanks for your suggestions.

(3) In methods, the fruit set and deformed rates are not clearly defined (lines 145-148). In addition, it is difficult to verify whether the fruit set of the pear was controlled for  the initial fruit before the fall of the initial fruit set or for the ripe harvested fruits.  In the case of hand pollination, the tested tree branches were not covered with a net, so the results were the effect of hand pollination and open pollination. 

Response: The number of flowers, fruit, and malformed fruit were monitored and recorded to calculate the fruit set rate (%) given by the number of fruits on the tomato plant / the number of flowers, and the malformed fruit rate (%) given by the number of malformed fruit / the number of fruits. Have provided this information in the revised manuscript.

Indeed, as you said, in the case of hand pollination and bumblebee pollination, the tested tree branches were not covered with a net, so the results were the synergic effect with open pollination. We acknowledged this. More ingenious experiments and observations should be conducted to ensure the pollination effects exactly in future research. We discussed this in the discussion section. Thanks for your suggestions.

(4) What cultivar was the pollen used for hand pollination?  

Response: The pollen of pear variety “Huangguan were used for hand pollination. Have provided this information in the revised manuscript. Thanks for your suggestions.

(5) If the orchard was a single variety (bumblebee pollination), without pollinizer and the pollen used in the hand pollination was another variety, it is difficult to compare pollination methods.  

Response: The pear varieties “Qiuyue” and “Huangguan” were planted in each of the nine pear orchards for seven years. Each orchard included 80 “Qiuyue” pear plants, and ten “Huang guan” pear variety as pollinizers. Have provided this information in the revised manuscript. Thanks for your suggestions.

(6) Was the distribution of variables tested before selecting statistical parametric tests?  On the other hand, the number of observations for the total yield (3 per group) was too small to compare averages. What was treated as an observation when the average percentage of fruit set was compared? Greenhouse/orchard or plant/tree? 

Response:Yes, all tests were conducted using SAS version 9.0 after verifying the normality. Yes, as you said, we acknowledged that the number of observations for the total yield (3 per group) was too small. More observations should be conducted to ensure the yield effect in future research. We discussed this in the revised manuscript. Fruit set of selected plants/trees were treated as an observation when the average percentage of fruit set was compared. We provided these information in the revised manuscript. Thanks for your suggestions.

(7) Line 374. Plant flowers do not secrete honey but nectar, however, bumblebees primarily need pollen. 

Response: Yes, plant flowers do not secrete honey but nectar. We corrected this according to your suggestions in the revised manuscript. Thanks for your suggestions.

(8) When interpreting results (e.g., lines 366-381), it is important to distinguish between facts that apply to populations of wild bumblebees vs. reared bumblebees. In the discussion, the authors should make a clearer distinction between the use of bumblebees in the greenhouse versus in the field. Also, nothing mentioned about the relationship of commercial bumblebee colonies vs. wild bumblebees and wild bees. As well as there is a lack of interpretation of pollination results and reference to literature data. 

Response: We distinguished between facts that apply to populations of wild bumblebees vs. reared bumblebees in the revised manuscript, and made a clearer distinction between the use of bumblebees in the greenhouse versus in the field in the revised manuscript. We also discussed the potential relationship of commercial bumblebee colonies vs. wild bumblebees and wild bees. Thanks for your suggestions.

(9) The reference list is double-numbered.

Response: Done.Thanks for your suggestions.

Reviewer 2 Report

Comments and Suggestions for Authors

The manuscript is about the significance of bumble bees, its be behavior and related production of Cherry Tomato and Pear Varieties in Northern China. The manuscript has been well written and authors followed the standard methodology for the better results. I have some suggestions for improvement of the manuscript before publication.

Comments to the authors

The manuscript is about the significance of bumble bees, its be behavior and related production of Cherry Tomato and Pear Varieties in Northern China. The manuscript has been well written and authors followed the standard methodology for the better results. I have some suggestions for improvement of the manuscript before publication.

Title

May be changed as below

Bumblebee Foraging Dynamics and Pollination Outcomes for Cherry Tomato and Pear Varieties in Northern China

Abstract

The abstract is well-structured, presenting key findings related to temperature-dependent foraging activities of B. terrestris, particularly in winter greenhouses and early spring pear orchards. It demonstrates the positive impact of B. terrestris pollination on cherry tomato and pear crops, with enhanced fruit yields, setting rates, weight, and seed numbers. Notably, the study compares the performance of B. terrestris pollination with 2,4-dichlorophenoxyacetic acid (2,4-D) treatments and other natural pollination methods.

To enhance the abstract, it would be beneficial to include a brief mention of the practical implications of the findings, such as potential economic benefits for farmers, sustainability improvements, or broader ecological considerations.

1.      Line 12 please you may add few more lines for methodology followed.

2.      Study gaps are missing in this section.

3.      Add two lines for future research based on your results.

Introduction

1.       Line 31-33 may be changed as below

Since Sladen's pioneering proposal of bumblebee pollination application in crops in 1912, as documented in "The Bumblebee," the utilization of bumblebees as pollinators has garnered substantial attention [1-3].

2.      Line 67-72 is suggested to change as below

CO2 can disrupt bumblebee diapause marked a pivotal moment in overcoming diapause challenges, offering crucial technical support for the artificial mass breeding of bumblebees [27]. Subsequently, companies like Biobest in Belgium and Koppert and Bunting Brinkman Bees (BBB) in the Netherlands initiated bumblebee breeding and global popularization efforts [13,19]. Today, the utilization of Bombus terrestris for pollination has become a global practice, gaining recognition for its pivotal role.

3.      Line 88-100 may be changed as below

The underutilization of bumblebee pollination in northern China is primarily attributed to farmers' lack of understanding of this technology and the existing knowledge gap regarding the foraging behavior and pollination effects of bumblebees on economically important crops in this region. To address this gap and facilitate the broader application of bumblebee pollination, comprehensive field studies comparing bumblebee pollination with other methods are essential. In this study, we aimed to evaluate the foraging behavior and pollination effects of Bombus terrestris on cherry tomato (Solanum lycopersicum L.) and pear (Pyrus spp.). Our objective is to provide empirical data and a theoretical foundation for selecting bumblebee pollination or alternative methods in the cultivation of economically important crops, thereby contributing to enhanced crop management in northern China.

As a whole

The introduction provides a historical context for the application of bumblebee pollination, emphasizing its significance in crop production globally. It discusses the essential role of bumblebees in collecting pollen and nectar for their survival and highlights their potential to enhance pollination in plants. The introduction also recognizes the widespread use of Bombus terrestris in developed countries and the economic benefits associated with its application in pollinating greenhouse crops.

The introduction then delves into historical milestones, such as the discovery that CO2 can break the diapause of bumblebees, leading to advancements in artificial mass breeding. The global recognition and use of Bombus terrestris for pollination are highlighted, contributing to the acknowledgment of its importance.

The narrative smoothly transitions to the current situation in China, emphasizing the limited application of bumblebee pollination technology and the prevailing reliance on traditional methods, including synthetic hormones, for improving fruiting rates in cherry tomatoes and hand-pollination for pear plants. The introduction effectively raises the crucial issue of the lack of understanding and knowledge gap among farmers in northern China regarding bumblebee pollination technology.

The final section sets the stage for the study's objectives, emphasizing the need for comparative field studies to evaluate the foraging behavior and pollination effects of bumblebees on cherry tomato and pear crops. The overall structure and flow of the introduction are well-organized, providing a clear context for the study and a rationale for the research questions addressed. Additionally, the inclusion of specific crops and the focus on northern China adds relevance and applicability to the study.

2. Materials and methods

Models, made and specifications of the instruments used in the study may be included. Most of the sections of this part lack proper citations for methodology followed.

1.      Line 103-105 please add citations to these lines.

2.      Line 197-201 need the addition of citations.

Additionally follow below suggestions

Clarify Experimental Design:

Provide a brief rationale for choosing Fei County in Shandong province as the research site. Explain if there are specific characteristics that make this location suitable for the study.

Clarify why the same variety of cherry tomatoes (Fushan-88) was chosen for the three-year greenhouse experiment. Is this variety commonly grown in the region?

Expand on Bumblebee Treatment:

Elaborate on the rationale for choosing the hormone 2,4-D as a comparison to bumblebee treatment. Are there specific challenges or trends related to hormone usage in northern China that prompted this comparison?

Provide more details on the foraging behavior observations. How were the observations conducted, and what specific behaviors were monitored in the bumblebee-treated greenhouses?

Enhance Clarity in Greenhouse Experiment:

Specify the duration of the flowering phase in cherry tomato plants. This information is crucial for understanding the timing of bumblebee and hormone treatments.

Clarify if there were any specific measures taken to standardize the application of 2,4-D to ensure consistency across treatments.

Provide Beehive Details:

Offer additional information about the purchased beehives, such as the criteria for selecting a healthy bumblebee population and any measures taken to maintain their health during the experiment.

Temperature and Humidity Monitoring:

Discuss the relevance of monitoring temperature and relative humidity during the bumblebee releasing period. How do these environmental factors impact bumblebee behavior, and what insights were gained from this monitoring?

Elaborate on Pear Orchard Experiment:

Justify the choice of pear variety (Qiuyue) and explain why it was selected for the orchard experiment. Provide more details on the methods of artificial pollination, especially how farmers conducted hand-pollination using brushes.

Improve Clarity in Data Analysis:

Elaborate on why the Student’s t-test and one-way analysis of variance (ANOVA) with Tukey’s HSD were chosen for data analysis.

3.      Results

I have below given suggestions for improvement of this section

Clarify Units and Symbols:

Ensure consistency in the use of temperature units (℃), and consider adding the corresponding symbols for temperature and relative humidity in the text to enhance clarity.

Improve Figure Labels:

In Figures 1 and 2, add clear labels to the axes, including units for temperature and relative humidity, and provide a brief title or caption to summarize the main findings.

Enhance Data Presentation:

Consider presenting temperature and relative humidity data in tabular form alongside the figures to provide a numerical reference for readers.

Improve Description of Results:

Provide more interpretation of the results. Explain the significance of temperature and humidity fluctuations on bumblebee foraging activities and how these environmental factors might influence pollination.

Highlight Peaks in Foraging Activities:

Emphasize the significance of the peak periods in bumblebee foraging activities in both greenhouses and pear orchards. Discuss potential reasons for these peaks and their relevance to pollination.

Consider a Subsection on Foraging Behavior:

Create a separate subsection specifically for the foraging behavior of bumblebees, including details on how the observations were conducted and any notable behaviors observed.

Improve Table Presentation:

Enhance the formatting of Table 1 by providing more space between rows and columns for better readability.

Provide Context for Bumblebee Behaviors:

Offer a brief discussion or literature review section to provide context for the observed foraging behaviors. How do the observed behaviors align with or differ from existing knowledge in the field?

Include Statistical Significance in Figures:

Indicate statistical significance directly on the figures, such as placing asterisks or symbols to highlight differences. This will make it easier for readers to quickly understand the significance of the results.

Enhance Figure Legends:

Expand the figure legends to provide more information on the significance of the findings, key observations, or trend.

4.      Discussion

This section has been written well from my side.

Author Response

Mar. 14, 2024

Dear reviewer,

We now return the revised edition of Manuscript ID insects-2893366, titled "Bumblebee foraging dynamics and pollination outcomes for cherry tomato and pear varieties in northern China".

We carefully revised the manuscript according to your comments, and addressed each comment (please see below), and believe that the manuscript has been improved.

We are grateful to your constructive comments on earlier drafts of this manuscript.

Thanks

With kind regards,

Yueyue Wang

Below we address each your comments:     

(1) Title may be changed as below “Bumblebee foraging dynamics and pollination outcomes for cherry tomato and pear varieties in northern China”.

Response: We changed the title according to your suggestion. Thanks for your suggestions.

(2) To enhance the abstract, it would be beneficial to include a brief mention of the practical implications of the findings, such as potential economic benefits for farmers, sustainability improvements, or broader ecological considerations.

Response: As effective pollinator, B. terrestris pollination can improve cultivation efficiency, increase yield, produce more economic benefits. Moreover, it can also play an important role in regulating agricultural operations, restricting the use of chemical pesticides, reducing hormone residues, and ensuring the safety of agricultural products. We gave a brief mention of this in the section of abstract. Thanks for your suggestions.

(3) Line 12 please you may add few more lines for methodology followed.

Response: We gave a brief mention of this in the section of abstract. Thanks for your suggestions.

(4) Study gaps are missing in this section.

Response: Done. Thanks for your suggestions.

(5) Add two lines for future research based on your results.

Response: However, the risk of introducing commercially available bumblebees into orchards to local bumblebee species should be considered and evaluated in future research. We gave a brief mention of this in the section of abstract. Thanks for your suggestions.

  • Line 31-33 may be changed as below. Since Sladen's pioneering proposal of bumblebee pollination application in crops in 1912, as documented in "The Bumblebee," the utilization of bumblebees as pollinators has garnered substantial attention[1-3]..

Response: Done. Thanks for your suggestions.

(7) Line 67-72 is suggested to change as below CO2 can disrupt bumblebee diapause marked a pivotal moment in overcoming diapause challenges, offering crucial technical support for the artificial mass breeding of bumblebees [27]. Subsequently, companies like Biobest in Belgium and Koppert and Bunting Brinkman Bees (BBB) in the Netherlands initiated bumblebee breeding and global popularization efforts [13,19]. Today, the utilization of Bombus terrestris for pollination has become a global practice, gaining recognition for its pivotal role..

Response: Done. Thanks for your suggestions.

(8) Line 88-100 may be changed as below The underutilization of bumblebee pollination in northern China is primarily attributed to farmers' lack of understanding of this technology and the existing knowledge gap regarding the foraging behavior and pollination effects of bumblebees on economically important crops in this region. To address this gap and facilitate the broader application of bumblebee pollination, comprehensive field studies comparing bumblebee pollination with other methods are essential. In this study, we aimed to evaluate the foraging behavior and pollination effects of Bombus terrestris on cherry tomato (Solanum lycopersicum L.) and pear (Pyrus spp.). Our objective is to provide empirical data and a theoretical foundation for selecting bumblebee pollination or alternative methods in the cultivation of economically important crops, thereby contributing to enhanced crop management in northern China..

Response: Done. Thanks for your suggestions.

(9) Line 103-105 please add citations to these lines.

Response: Done. Thanks for your suggestions.

(10) Provide a brief rationale for choosing Fei County in Shandong province as the research site. Explain if there are specific characteristics that make this location suitable for the study.

Response: The research was conducted in Fei (35.318°N, 118.045°E), a large agricultural county, in Shandong province of China. The mean annual temperature in the study area is 13.6 °C. Air temperatures can fall to - 11 °C in December and reach 36 °C in July. The mean annual precipitation is 850 mm. Tomatoes and pears are important economic crops in this area. The annual tomato and pear production is up to ~270,000 tons and ~45,605 tons, worth USD $280 million and USD $23 million, respectively (http://nyj.linyi.gov.cn/). To improve tomato production yields in greenhouses, the traditional method used most often was treating flowers by 2,4-D. For pollination of pear plants in orchards, farmers generally hand-pollinate using brushes, and even adopt open pollination. Bumblebee pollination technology is still applied less. We supplied this information in the revised manuscript. Thanks for your suggestions.

(11) Clarify why the same variety of cherry tomatoes (Fushan-88) was chosen for the three-year greenhouse experiment. Is this variety commonly grown in the region?

Response: Cherry tomato variety (Fushan-88) is the commonly cultivated in this region. We supplied this information in the revised manuscript. Thanks for your suggestions.

(12) Elaborate on the rationale for choosing the hormone 2,4-D as a comparison to bumblebee treatment. Are there specific challenges or trends related to hormone usage in northern China that prompted this comparison?

Response: To improve tomato fruiting rate and production yields in greenhouses, the traditional method used most often was treating flowers using synthetic hormones 2,4-D.  However, the excessive application of hormone or its residues can cause poisoning in humans or have adverse effects on the environment [31-33]. And, the use of synthetic hormones treating flowers are expensive in terms of both labor and time. These specific challenges prompted this comparison. Have introduced this in the introduction section. Thanks for your suggestions.

(13) Provide more details on the foraging behavior observations. How were the observations conducted, and what specific behaviors were monitored in the bumblebee-treated greenhouses?

Response: We monitored bumblebee foraging activities from 9:00 to 17:00 in each greenhouse with bumblebee release on 29 November 2020. In each greenhouse, we made hourly observations and recorded the number of bees leaving the bumblebee hive within 10 min (individuals per 10 min) and the number of returning bees carrying pollen within 10 min (individuals per 10 min) from 9:00 to 17:00. At the same time, ten bumblebees leaving the bumblebee hive were randomly selected to observe the number of visited flowers per min per individual and the single flower residence time (s) of bumblebee during the period from 12:00 -13:00. Have introduced this in the revised manuscript. Thanks for your suggestions.

(14) Specify the duration of the flowering phase in cherry tomato plants. This information is crucial for understanding the timing of bumblebee and hormone treatments.

Response: The duration of the flowering phase was about 45 days. We supplied this information in the revised manuscript. Thanks for your suggestions.

(15) Clarify if there were any specific measures taken to standardize the application of 2,4-D to ensure consistency across treatments.

Response: During the flowering period, we used 5 mg × L -1 solution of 2, 4-D to treat tomato inflorescence. About 1 mL of solution was used for each inflorescence. Have introduced this. Thanks for your suggestions.

(16) Offer additional information about the purchased beehives, such as the criteria for selecting a healthy bumblebee population and any measures taken to maintain their health during the experiment.

Response: Before installing into the greenhouse all three hives were kept in laboratory room (temperature 26-28℃; relative humidity 60-70%) for 24 hours. All hives were comprehensively checked for health conditions and presence of any pest or predator. And, each hive was free from diseases and microbes. Bumblebee queens were with glistening color without any damage in their body and actively attending the whole colony. About 100 active bumblebee workers with good conditioned wings were present in the colony along founder queen. Each hive box held a plastic tank in the base of box that stored a sugar solution as substitute for nectar to sustain the nutritional development of bumblebee’s health as tomato flowers do not produce nectar. With the interval of 7 days, continually examination of colonies was made to check the population conditions and presence of pest or predators in the hive. Dead bumblebee workers were removed from the hive to avoid contamination. We supplied this information in the revised manuscript. Thanks for your suggestions.

(17) Discuss the relevance of monitoring temperature and relative humidity during the bumblebee releasing period. How do these environmental factors impact bumblebee behavior, and what insights were gained from this monitoring?

Response: We have discussed this in the Discussion section. Thanks for your suggestions.

(18) Justify the choice of pear variety (Qiuyue) and explain why it was selected for the orchard experiment. Provide more details on the methods of artificial pollination, especially how farmers conducted hand-pollination using brushes.

Response: “Qiuyue” is the commonly cultivated pear variety in this area. Pear flowers began to bloom in early April of each year. At this time, the bumblebees were released to the pear plants in the three of the nine orchards. One bumblebee hive was placed in center of each orchard, until the end of flowering phase. The bumblebee pollination of pear flowers lasted ~14 days in each orchard. The condition and maintenance of each bumblebee hive was the same as above. Another three orchards were randomly selected to adopt hand pollination. Here, the farmers hand-pollinated flowers using brushes dipping with pollen to touch the pear flowers during the flowering phase in early April. The pollen of pear variety “Huangguan” were used for hand pollination. The remaining three orchards adopted open pollination, with no hand pollination or bumblebees introduced to the area. Have introduced this in the revised manuscript. Thanks for your suggestions.

(19) Ensure consistency in the use of temperature units (℃), and consider adding the corresponding symbols for temperature and relative humidity in the text to enhance clarity.

Response: Done. Thanks for your suggestions.

(20) In Figures 1 and 2, add clear labels to the axes, including units for temperature and relative humidity, and provide a brief title or caption to summarize the main findings.

Response: Done. Thanks for your suggestions.

(21) Provide more interpretation of the results. Explain the significance of temperature and humidity fluctuations on bumblebee foraging activities and how these environmental factors might influence pollination.

Response: Yes, we provided more interpretation, please see the Discussion section. Thanks for your suggestions.

(22) Emphasize the significance of the peak periods in bumblebee foraging activities in both greenhouses and pear orchards. Discuss potential reasons for these peaks and their relevance to pollination.

Response: Done. Please see the Discussion section. Thanks for your suggestions.

(23) Enhance the formatting of Table 1 by providing more space between rows and columns for better readability.

Response: Done. Thanks for your suggestions.

(24) Offer a brief discussion or literature review section to provide context for the observed foraging behaviors. How do the observed behaviors align with or differ from existing knowledge in the field?

Response: Done. Please see the Discussion section. Thanks for your suggestions.

(25) Indicate statistical significance directly on the figures, such as placing asterisks or symbols to highlight differences. This will make it easier for readers to quickly understand the significance of the results.

Response: Done. Thanks for your suggestions.

Reviewer 3 Report

Comments and Suggestions for Authors

The manuscript is well-written and easily understandable, in combination with the topic it is likely appealing to the readership of Insects. While the overall quality of the manuscript is commendable, some minor improvements in English language usage are warranted to enhance clarity and coherence.

However, there is a serious concern regarding the conclusion, particularly as the risks associated with introducing reared bumblebees into natural environments (desease transmission, changes in local gene pool, etc.) for pollination purposes are not discussed at all. Instead, the authors "recommend its wide application to [...] pear cultivation in orchards" already in the abstract. This should definitely be discussed and, if at all, stated with more caution.

Major Comments:

1) The authors should discuss the (environmental) risks of introducing commercially available bumblebees into orchards and not just make an unqualified recommendation for their use. Particularly as it is recently discussed that an introduction of Bombus terrestris to e.g. China is highly problematic to local bumblebee species and their pollination potential, see e.g.
- Orr et al, Entomologia Generalis. 2022, Vol. 42 Issue 4, p655-658. 4p. ;
and for other regions e.g.
- Whitehorn et al, Journal of Apicultural Research 2016, 52:3, 149-157,
- Fürst et al., 
Nature 506, 364–366 (2014)
- Otterstatter and Thomson, PLoS ONE 2023,
https://doi.org/10.1371/journal.pone.0002771

2) In the experiments on pear orchids, information is missing, who the natural pollinators are and how the interactions/effects between them and the introduced bumblebees are. Is the pollination success additive or exclusive between these two groups? How can you tell the effects apart?

Minor Comments:

- The literature cited does not always support the statements made and other statements are not backed up by references at all. All references should be checked for relevance and missing references shoud be provided were applicable; just a few examples:

o   lines 41-45: the references [10-12, 14, 15] are about general ecological traits (e.g. chemical ecology, sesnse of taste, etc.) of bumblebees and do not support the statement in the text which is about their economic value and their widely use in agricultural production systems in Europe

o   lines 55-57: reference [22] is about comparison of (partially manual) pollination methods and not about „behavioral characteristics of bumblebees“ as stated in the respective paragraph

o   line 57-58: no references are cited for the statement „[bumblebees] visit flowers when the temperature is 8 °C, while honeybees generally work above 16 °C“

o   line 59: the cited literature [24] is about the interaction of temperature and sucrose concentration on decicion making in bumblebees, and not about sonication, which is the point made in the respective line

o   line 332-335: reference [1] is about global trends and threats of bumblebee populations, while the paragraph is about factors influencing foraging behavior of bumble bees

o   line 335-338: reference [35] is about warmth as a floral trait (with color only discussed as a predictor of warmth), but temperature is not discussed in the paragraph as a floral trait

-  Introduction: what is the situation of natural bumble bee species / populations in China, how can this be improved e.g. for pollination purposes in pear orchards

-  line 57: why is „weak phototaxis“ liste das a benefitial trait of bumblebees for use in greenhouse pollination? This needs some explanation.

-  Line 59: „loud sound“ is not the right term, probably „buzz pollination“ or „sonication“ is meant here

-  Lines 63-64: the statement gives the impression that bumblebees leave their nest in the morning and do not return until dusk; that’s not true as foragers frequently return to the hive to deposit pollen and (nectar) loads

- Discussion part: see Major Comment 1)

Author Response

Mar. 14, 2024

Dear reviewer,

We now return the revised edition of Manuscript ID insects-2893366, titled "Bumblebee foraging dynamics and pollination outcomes for cherry tomato and pear varieties in northern China".

We carefully revised the manuscript according to your comments, and addressed each comment (please see below), and believe that the manuscript has been improved.

We are grateful to your constructive comments on earlier drafts of this manuscript.

Thanks

With kind regards,

Yueyue Wang

Below we address each your comments:     

(1) The authors should discuss the (environmental) risks of introducing commercially available bumblebees into orchards and not just make an unqualified recommendation for their use. Particularly as it is recently discussed that an introduction of Bombus terrestris to e.g. China is highly problematic to local bumblebee species and their pollination potential.

Response: Indeed, the risk of introducing commercially available bumblebees into orchards in northern China should be considered and evaluated cautiously. The global success of B. terrestris as both a commercial pollinator and an invasive species has had unfortunate impacts. This species has negatively impacted other bumblebees, even leading to local extirpations of native competitors as it continues to spread [54-55]. In addition, researchers has highlighted the reared Bombus terrestris as risk factor for wild bumblebees due to transfer of pathogen [56-58]. So there should be intense care about commercially available bumblebees to decrease risk of interaction of wild bumblebee populations. And, the domestication and artificial mass breeding of native bumblebees for pollination are the best approaches to avoid the risk of introducing non-native bumblebees. We discussed the potential risk of introducing commercial bumblebee colonies in the Discussion section. Thanks for your suggestions.

(2) In the experiments on pear orchids, information is missing, who the natural pollinators are and how the interactions/effects between them and the introduced bumblebees are. Is the pollination success additive or exclusive between these two groups? How can you tell the effects apart?

Response: For open pollination, we observed that the natural pollinators mainly included insects occurring in early spring, such as some wild honeybees. Indeed, as you said, we also acknowledged that, in the case of hand pollination and bumblebee pollination in the present study, the tested tree branches of pear were not covered with a net, so the results were the synergic effect with open pollination. More ingenious experiments and observations should be conducted to ensure the yield effects exactly in future research. We discussed this in the Discussion section. Thanks for your suggestions. 

(3) The literature cited does not always support the statements made and other statements are not backed up by references at all. All references should be checked for relevance and missing references shoud be provided were applicable.

Response: So sorry for the wrong arrangement of these references. All references were checked again for relevance and then corrected according to your suggestions in the revised manuscript. Thanks for your suggestions.

(4) Introduction: what is the situation of natural bumble bee species / populations in China, how can this be improved e.g. for pollination purposes in pear orchards

Response: China is a hotspot for high bumblebee species richness. There are about 125 extant species, representing 14 of the 15 Bombus subgenera (Sun et al., 2020). However, the artificial mass breeding of those native bumblebees have not succeeded. Hence, the  reared populations are still not applied for the pollination of economic crops. We provided these information in the introduction section in the revised manuscript. Thanks for your suggestions.

(5) line 57: why is, weak phototaxis“ liste das a benefitial trait of bumblebees for use in greenhouse pollination? This needs some explanation.

Response: We deleted this wrong description in the revised manuscript. Thanks for your suggestions.

(6) Line 59: loud sound“ is not the right term, probably „buzz pollination“ or „sonication“ is meant here

Response: We changed to “buzz pollination” according to your suggestions. Thanks for your suggestions.

(7) Lines 63-64: the statement gives the impression that bumblebees leave their nest in the morning and do not return until dusk; that’s not true as foragers frequently return to the hive to deposit pollen and (nectar) loads

Response: We changed this into “In summer, some worker bees forage at 5:00 AM, and stop working until dark”. Thanks for your suggestions.

(8) Discussion part: see Major Comment 1).

Response: The risk of introducing commercially available bumblebees into orchards to local bumblebee species should be considered and evaluated in future research. We discussed this in the Discussion section. Thanks for your suggestions.

Round 2

Reviewer 3 Report

Comments and Suggestions for Authors

The authors have responded satisfactorily to all comments and have made appropriate changes to the text. The quality of the manuscript has improved considerably as a result, so that I have no further reservations regarding its publication.